# A Novel Zn_2_Cys_6_ Transcription Factor, TopC, Positively Regulates Trichodin A and Asperpyridone A Biosynthesis in *Tolypocladium ophioglossoides*

**DOI:** 10.3390/microorganisms11102578

**Published:** 2023-10-17

**Authors:** Xiang Liu, Rui-Qi Li, Qing-Xin Zeng, Yong-Quan Li, Xin-Ai Chen

**Affiliations:** 1School of Medicine and the Children’s Hospital, Zhejiang University, Hangzhou 310058, China; 22018007@zju.edu.cn (X.L.); 21918020@zju.edu.cn (R.-Q.L.); 2Institute of Pharmaceutical Biotechnology, Zhejiang University, Hangzhou 310058, China; lyq@zju.edu.cn; 3Sir Run Run Shaw Hospital, Zhejiang University School of Medicine, Hangzhou 310058, China; 3415018@zju.edu.cn

**Keywords:** asperpyridone A, trichodin A, biosynthesis, TopC, *Tolypocladium ophioglossoides*, cryptic gene cluster, Zn_2_Cys_6_ transcription factor

## Abstract

Asperpyridone A represents an unusual class of pyridone alkaloids with demonstrated potential for hypoglycemic activity, primarily by promoting glucose consumption in HepG2 cells. Trichodin A, initially isolated from the marine fungus Trichoderma sp. strain MF106, exhibits notable antibiotic activities against *Staphylococcus epidermidis*. Despite their pharmacological significance, the regulatory mechanisms governing their biosynthesis have remained elusive. In this investigation, we initiated the activation of a latent gene cluster, denoted as “*top*”, through the overexpression of the Zn_2_Cys_6_ transcription factor TopC in *Tolypocladium ophioglossoides*. The activation of the *top* cluster led to the biosynthesis of asperpyridone A, pyridoxatin, and trichodin A. Our study also elucidated that the regulator TopC exerts precise control over the biosynthesis of asperpyridone A and trichodin A through the detection of protein–nucleic acid interactions. Moreover, by complementing these findings with gene deletions involving *topA* and *topH*, we proposed a comprehensive biosynthesis pathway for asperpyridone A and trichodin A in *T. ophioglossoides*.

## 1. Introduction

Filamentous fungi are important producers of structurally diverse secondary metabolites and a huge source of discovering novel and commercially important pharmaceuticals, mycoinsecticides, and antibiotics [1]. Secondary metabolites synthesized by fungi include polyketide, non-ribosomal peptide, terpene, hybrid polyketide-nonribosomal peptide, and alkaloids [2,3,4]. Several potential fungus-based drugs, such as penicillin, echinocandins [5], cyclosporines [6], and lovastatin [7], play crucial roles in the field of therapeutic and biomedical sciences.

With the development of genome sequencing and bioinformatic analyses, numerous biosynthetic gene clusters have been identified within fungal genomes. This suggests that fungi possess the potential to produce a much wider array of natural products than previously anticipated [8,9]. However, most clusters are cryptic or have low expressions under laboratory conditions, preventing the discovery of their associated products. Various activation strategies have been employed to awaken these silent gene clusters, enabling the production of natural products with novel structures and potential pharmaceutical applications. These strategies include optimizing culture conditions, heterologous expression, cocultivation, overexpressing pathway-specific positive regulatory factors, or global regulators, and replacing promoters, among others [10,11,12].

*Tolypocladium ophioglossoides* are known to produce various secondary metabolites (SMs), including tyrosol, cordyepolA-C, ophiosetin, and balanol [13,14]. There are 31 gene clusters in the genome of *T. ophioglossoides* [13], indicating its potential for synthesizing a wide range of natural products. However, transcriptome data indicate that most of the gene clusters exhibit low expression levels under laboratory culture conditions. It is intriguing to consider activating these latent gene clusters using appropriate strategies.

The 4-hydroxy pyridones represent a class of polyketide-nonribosomal peptide hybrid compounds with diverse heterocyclic structures and bioactivities [15]. They have been isolated from various sources, including *Aspergillus terreus* [16], *A. flavus* [17], *A. nidulans* [18], and *Cylindrocladium ilicicola* [19]. This group of compounds includes flavipucines, leporins, aspyridone A, ilicicolin H, and several others. The biosynthesis mechanism of 4-hydroxy pyridones has been extensively studied over the past three decades. Tenellin is the first 4-hydroxy pyridine for which the biosynthesis has been distinctly elucidated [20]. The biosynthesis of 4-hydroxy pyridones originated from the condensation of acetyl-CoA, malonyl CoA, and tyrosine, yielding an intermediate pentacyclic structure known as acryltetramic acid. This transformation is catalyzed via a polyketide synthase–nonribosomal peptide synthetase (PKS-NRPS) and a trans-acting enoyl reductase (ER) [15]. Some PKS-NRPSs exhibit a broader substrate tolerance, exemplified by ApdA and TolA, enabling to the utilization of alternative aromatic amino acids and producing analogs [21]. Subsequently, the pentacyclic structure of acryltetramic acid was transformed into the six-membered 4-hydroxy-2-pyridone catalyzed via a cytochrome P450 enzyme [20]. The compounds containing 4-hydroxy-2-pyridone undergo modification using various postmodification enzymes, including intramolecular Diels–Alder reactions [22], Alder-ene reaction [23], epimerization [19], hydroxylation [22], and so on.

Pyridoxatin and its derivatives contain the 4-hydroxy-3-alkyl pyridones structure, which has been isolated from *Acremonium* sp (BX86) [24], *Chaunopycnis* sp (CMB-MF028) [25], and *A. bombycis* [23]. Bioactive investigations have shown that pyridoxatin and its derivatives exhibit antimalarial [26] and antibiotic activity [27,28], as well as free radical scavenger. Asperpyridone A, a potential hypoglycemic agent, was first isolated from the endophytic fungus *Aspergillus* sp. TJ2329 [29].

In fungi, the pathway-specific activator is typically localized within the gene cluster, and its overexpression using a strong promoter or knockout is a simple and efficient strategy to activate the cryptic gene cluster. The fungal transcription factor GAL4 contains six highly conserved cysteine residues, forming the motif CX_2_CX_6_CX_6_CX_2_CX_5_CX_2_, commonly referred to as the C6 transcription factor. The GAL4-like Zn_2_Cys_6_ binuclear cluster DNA-binding domain is exclusively found in fungi. Transcription factors containing this domain also include STB5, DAL81, CAT8, RDR1, and HAL9 in *Saccharomyces cerevisiae*. These transcription factors typically play a regulatory role in numerous secondary metabolic biosyntheses. For instance, overexpression of the transcription factor ApdR in *A. nidulans* has been shown to induce the production of aspyridones A and B [30]. The MokH transcription factor, containing the Zn(II)_2_Cys_6_ binuclear DNA binding domain, serves as an activator for monacolin K production.

In this study, we activated a cryptic gene cluster (located on chromosome 4, region 1) via the overexpression of one fungal-specific positive activator TopC in *T. ophioglossoides*. We identified this gene cluster as the *top* gene cluster which could simultaneously produce pyridoxatin, trichodin A, and asperpyridone A. We first ascertained that TopC positively regulates the transcription of relative genes by binding to promoters of these genes within the *top* cluster.

## 2. Materials and Methods

### 2.1. Strain and Culture Conditions

*T. ophioglossoides* and its transformants were cultivated on potato dextrose agar (PDA) or COB (5% polypeptone, 5% yeast extract, 1% MgSO_4_·7H_2_O, 0.5% KH_2_PO_4_, 3% sucrose, pH 5.5) to obtain the compounds at 26 °C. *E. coli* strains were grown in Luria-Bertani (LB) or on LB agar plates at 37 °C for 12 h. YEP medium (1% peptone, 1% yeast extract, 0.5% NaCl, pH 7.0) was used for the culture of *Agrobacterium tumefaciens* AGL1 with appropriate antibiotics if needed (Appendix A). 

### 2.2. T. ophioglossoides Genomic DNA Extraction

*T. ophioglossoides* was cultured in COB medium for a duration of 4 days. Subsequently, the filamentous mycelium was harvested using miracloth and thoroughly rinsed with ultrapure sterile water. Genomic DNA was then extracted from the mycelia. Following this, the precipitate and supernatant were effectively separated through centrifugation at a rate of 12,000 rpm for approximately 5 min. A volume of 500 μL of supernatant was carefully transferred to a fresh Eppendorf tube, to which an equivalent volume of isopropanol was promptly added. This mixture was left to stand at room temperature for a period of 10 min. Subsequently, the supernatant was decanted, and the sediment underwent two additional washes with 75% ethanol.

### 2.3. Construction and Cloning of Fungal Recombinant Plasmids

Genomic DNA from *T. ophioglossoides* was prepared as described. Primers were synthesized using GENERAY (Hangzhou, China). The plasmid pFGL-815N, a shuttle expression vector compatible with both *E. coli* and *A. tumefaciens*, was utilized. The sur resistance gene fragment was obtained via PCR amplification with primers *sur*-F/R. This fragment was subsequently inserted into the shuttle plasmid pFG-815N (digestion with restriction enzymes *EcoR* I and *Kpn* I) using cloning enzymes, yielding the plasmid pFG-SUR. The TEF promoter for the overexpression gene, derived from the translation elongation factor *tef* gene, is amplified via PCR using primers pTEF-F/R. The terminator of the T-*amy*B gene, which serves as the terminator for transcription activation factors, is amplified using primers T-*amy*B-F/R. The two promoters and terminators were then integrated into the constructed pFG-SUR plasmid (digestion with *Kpn* I), leading to the formation of the overexpression of plasmid pTEFTAS-2P+2T.

The *topC* gene is amplified using primers *topC*-p-F/R from *T. ophioglossoides* genomic DNA. The amplified *topC* gene is then cloned into the pTEFTAS-2P+2T plasmid, which is digested with *Spe I*, resulting in the shuttle plasmid pTEFTAS-*topC* overexpression. The primers and plasmids used in this procedure are listed in Appendix A and Appendix A.

### 2.4. Transformation of T. ophioglossoides

*T. ophioglossoides* transformation was performed via the ATMT method as described previously [31].

The transformants of *T. ophioglossoides* need to be subcultured for three to four generations to form homozygous mutants. A list of all strains constructed in this study is provided in Appendix A.

### 2.5. High Performance Liquid Chromatography (HPLC) Analysis and Liquid Chromatog Raphy-Mass Spectrometry (LC-MS) Analysis of Secondary Metabolites

The transformant strains were transferred into 100 mL COB medium in a 250 mL flask and cultured at 26 °C and 180 rpm for 14 days. Then, the culture broth and mycelia were separated via filtration. The culture broth was extracted three times with an equal volume of ethyl acetate and the extracts were analyzed to determine metabolite changes via HPLC. The mycelia were disrupted two times with methanol under ultrasonic conditions. The extracts of the mycelia were directly analyzed using HPLC between 200 and 640 nm.

HPLC analysis was carried out using an Agilent 1260 Infinity system with a DAD detector and a reversed-phase C18 column (Agilent Eclipse Plus C18, 4.6 × 250 mm, 5 µm, Agilent Technologies, Hangzhou, China). Chromatographic conditions were as follows: solvents: (A) water + 1% Formic acid (FA), and (B) acetonitrile; solvent gradient 5% B in the first 5 min, increased to 100% at 45 min, to 5% B at 46 min, followed by 4 min with 5% B. This runs at a flow rate of 1 mL/min, a column temperature of 40 °C and UV detection at 200 to 640 nm.

LC-MS analysis was conducted using an Agilent 1200 HPLC system (Agilent, Santa Clara, CA, USA) and a Thermo Finnigan LCQDeca XP Max LC/MS system (Thermo Finnigan, Waltham, MA, USA). The column employed was Agilent Eclipse Plus C18, while the mobile phases A and B consisted of H_2_O (with 0.1% formic acid) and acetonitrile (with 0.1% formic acid), respectively. The analysis involved a linear gradient from 5% to 100% (*v*/*v*) B over a duration of 50 min.

### 2.6. Compound Purification

HPLC purification was performed using a semi-preparative reverse-phase C18 column. 

A flow rate of 2.5 mL/min was employed with a solvent gradient system consisting of acetonitrile and water containing 0.1% formic acid. Absorbance was continuously monitored at wavelengths of 280 nm and 210 nm. The purification of compounds was carried out as follows: Compound **1**: purified via semipreparative HPLC using ACN-H_2_O (0.1% formic acid) in a ratio of 59:41 *v/v* as the mobile phase, resulting in a yield of **1** (1.5 mg, 36 min). Compound **2**: purified via semipreparative HPLC using ACN-H_2_O (0.1% formic acid) in a ratio of 52.5:47.5 *v/v* as the mobile phase, resulting in a yield of **2** (160 mg, 28 min). Compound **3**: purified via semipreparative HPLC using ACN-H_2_O (0.1% formic acid) in a ratio of 38.5:61.5 *v/v* as the mobile phase, resulting in a yield of **3** (2 mg, 58 min). Compound **4**: purified via semipreparative HPLC using ACN-H_2_O (0.1% formic acid) in a ratio of 52.5:47.5 *v/v* as the mobile phase, resulting in a yield of **4** (7 mg, 36 min). Compound **5**: purified via semipreparative HPLC using ACN-H_2_O (0.1% formic acid) in a ratio of 27.5:72.5 *v/v* as the mobile phase, resulting in a yield of **5** (8.5 mg, 34 min). Compounds **6a** and **6b**: purified via semipreparative HPLC using ACN-H_2_O (0.1% formic acid) in a ratio of 35:65 *v/v* as the mobile phase, resulting in yields of **6a** and **6b** (80 mg, 25 min). 

### 2.7. Quantitative Real-Time PCR (qRT-PCR) Analysis of Gene Expression

The mycelia of *T. ophioglossoides* were collected following 4 days of cultivation in COB liquid medium at 26 °C and 180 rpm. Total RNA extraction was performed using HiPure Total RNA Mini Kit (Magen, Guangzhou, China). Reverse transcription was conducted using HiScript III 1st Strand cDNA Synthesis Kit (Vazyme, Nanjing, China). The resulting cDNA was subjected to real-time quantitative PCR (qRT-PCR) for transcriptional analysis employing specific primers that generated PCR products of approximately 200 bp (Appendix A). qRT-PCR analysis of the cluster gene transcription was executed using Taq Pro Universal SYBR qPCR Master Mix (Vazyme, Nanjing, China). The *tef* gene, which encodes the translation elongation factor and maintains a constant expression level, was employed as an internal control [13]. The qRT-PCR reactions were initiated by incubating the samples at 95 °C for 1 min followed by 40 cycles at 95 °C for 15 s, 55 °C for 15 s, and 68 °C for 20 s. All samples were run in triplicate. The threshold cycle (Ct) was calculated from the program. The 2^−∆∆Ct^ method was used to quantify the relative changes in gene expression [31].

### 2.8. Heterologous Expression and Purification of TopC-DBD in E. coli

A DNA fragment encoding the TopC DNA binding domain (topC-DBD) was amplified from *T. ophioglossoides* cDNA and inserted into the pET-32a vector (digestion with *EcoR* I) via in-fusion cloning technology. The *E. coli* BL21 (DE3) cells harboring the expression plasmid were capable of producing the target TopC-DBD protein upon induction with 0.1 mM IPTG. The induced cells were incubated at 16 °C for 16 h. Affinity purification using Ni-agarose (Qiagen, Hilden, Germany) was employed to isolate the target TopC-DBD protein. The concentration of TopC-DBD was analyzed using a Bradford assay. 

### 2.9. Electrophoretic Mobility Shift Assay (EMSA)-Based Affinity Analysis

DNA probes corresponding to topAH, topB, topC, topDE, and topFG were generated by amplifying the respective genomic segments from *T. ophioglossoides*. Primers with 5′-end 6-carboxyfluorescein (6-FAM) labels were used for this purpose (Appendix A). The promoter of the *g8899* gene in the *T. ophioglossoides* genome was selected as a non-specific promoter for use as a blank control. As the negative control (Free vector probe), 1 μg of the promoter of g8899 was amplified from *T. ophioglossoides* genomic DNA using related primers. Detection of the FAM-labeled probes was carried out using the LAS4000 machine. The Electrophoretic Mobility Shift Assay (EMSA) was conducted to confirm the interaction between TopC-DBD and the DNA probe, as previously reported [32].

## 3. Results

### 3.1. Characterization of Pathway-Specific Regulator TopC of the Cryptic Top Cluster in T. ophioglossides

Based on the analysis of the genome sequence of *T. ophioglossoides* for all gene clusters, a gene cluster containing a PKS-NRPS hybrid enzyme was found. The transcriptome analysis showed that this gene cluster (*top*) was a cryptic gene cluster, with its constituent genes displaying low expression levels under our laboratory fermentation conditions (Appendix A). According to antiSMASH prediction, the *top* gene cluster, homologous to the *tol* gene cluster, encompasses a PKS-NRPS hybrid enzyme (*topE*) and various modified enzymes. These enzymes include one hypothetical methyltransferase (*topB*), a short-chain dehydrogenases/reductases (*topH*), three cytochrome p450 monooxygenases (*topA*, *F*, *G*), one enoyl reductase (*topD*), and one C6 transcription factor (*topC*), which are likely involved in the synthesis of complex natural products (Appendix A and Appendix A). 

Within this cluster, our investigation unveiled the presence of a putative activator gene named *topC*. This gene encodes a fungal cluster-specific C6 transcriptional factor. (Figure 1), which likely plays a role in regulating the expression of cluster genes. The full length of the TopC protein contains 802 amino acids and has a molecular weight of approximately 87.76 kDa. Bioinformatics analysis indicated that TopC is a typical multidrug-resistant transcription factor in fungi. It possesses an N-terminal CAL4-type Zn_2_Cys_6_ DNA-binding domain and a C-terminal fungal transcription factor regulatory middle homology domain (Figure 1 and Appendix A). 

To further explore the relationship of TopC, we conducted a phylogenetic analysis. We gathered nine Zn_2_Cys_6_ transcription factors exhibiting significant similarity to the TopC amino acid sequence. These factors were obtained from the NCBI protein database via a blast alignment search. Notable transcription factors in this group include yanR (Accession no. G3Y415.1) from *A. niger*, GAL4 (Accession no. QGN14419.1) from *Saccharomyces*, AflR (Accession no. P43651.3) from *A. parasiticus*, MdpE (Accession no. AN0148) from *A. nidulans* FGSC A4, AnTF (Accession no. AAC49195) from *A. nidulans*, MlcR (Accession no. Q8J0F2.1) from *P. citrinum*, MokH (Accession no. Q3S2U4.1) from *Monascus pilosus*, LovE (Accession no. Q0C8L8.1) from *A. terreus* NIH2624, and ApdR (Accession no. XP_045268485.1) from *Colletotrichum gloeosporioides*.

Our phylogenetic analysis suggests that TopC forms a distinct clade with *A. nidulan* ApdR, a recognized transcriptional activator with a pivotal role in aspyridone A and B biosynthesis in *A. nidulan* [33].

Given the prevailing role of C6 transcriptional factors as positive regulators, we devised an expression vector for the overexpression of *topC*. This vector drove the activation of the target gene cluster under the control of the strong constitutive promoter *pTEF*, a strategy previously documented [13]. Consequently, we generated the *topC*OE mutant. The identity of these transformants was confirmed through PCR analysis.

Notably, the *topC*OE mutants exhibited a significantly slower growth rate and displayed a more pronounced yellow pigmentation on PDA medium in comparison to the WT strains (Appendix A).

Examination of HPLC chromatograms from *topC*OE cultures revealed a markedly distinct metabolite profile compared to that of the WT strain. Six previously unobserved peaks (1–6) were detected and subsequently characterized within these new HPLC peaks (Figure 2). These findings strongly suggest the activation of the target gene cluster upon *topC* overexpression.

### 3.2. Structural Determination of Compounds ***1***–***6***

To characterize the chemical structures of these metabolites produced by the transformants, large-scale fermentation in liquid culture was carried out to isolate sufficient amounts of these compounds (Figure 3 and Appendix A).

Compound **1** was isolated as a light yellow amorphous solid. It has a molecular formula of C_21_H_27_NO_4_ based on the positive mode of HRESIMS. ^1^H-NMR data indicate that the structure of compound **1** is tolypoalbin, which could be isolated from *T. album* TAMA 479 [34] (Appendix A).

Compound **2** was isolated as a light yellow powder. It has a molecular formula of C_21_H_27_NO_5_ based on the negative mode of HRESIMS. ^1^H and ^13^C NMR data indicate that the structure of compound **2** is consistent with that of compound F-14329 isolated from *Chaunopycnis* sp. [25] (Appendix A and Appendix A).

Compound **3**, a white amorphous solid, has a molecular weight of 339.1898 (*m*/*z* 340.1898, [M + H]^+^) determined via HRESIMS, indicating the molecular formula of C_21_H_25_NO_3_. The UV, molecular formular, and ^1^H NMR data that indicate compound **3** have the same structure as trichodin A [28] (Appendix A and Appendix A). 

Compound **4** was isolated as colorless massive crystals, which have the molecular formular of C_16_H_23_NO_3_ via analysis of the HRESIMS spectrum. The UV and molecular formular data revealed that compound **4** has a similar skeleton structure to compound **6**. ^1^H and ^13^C NMR data of compound **4** were identical to those of asperpyridone A [29] (Appendix A and Appendix A).

Compound **5** was isolated as orange powder, of which the molecular formular was determined as C_21_H_27_NO_6_ via HRESIMS spectrum analysis. The UV and molecular formular data of **5** were similar to those of F-14329, indicating that compound **5** has a similar structure to compound **2**. Subsequently, the structure of compound **5** was determined via ^1^H and ^13^C NMR spectra, indicating that compound **5** was chaunolidine B (Appendix A and Appendix A).

Compounds **6a** and **6b**, light yellow powders, have a molecular weight of 263.1456 (*m/z* 262.1456, [M-H]^−^) determined via HRESIMS, indicating the molecular formula of C_15_H_21_NO_3_; the ratio of **6a** to **6b** is approximately 5:3 in DMSO (Appendix A and Appendix A). The structure of compounds **6a** and **6b** was elucidated as pyridoxatin based on ^1^H and ^13^C NMR data, which is a free radical scavenger, and could inhibit lipid peroxidation induced by free radicals in rat liver microsomes free from vitamin E [24].

Pyridoxatin and trichodin A with asperpyridone A are the known unusual pyridone alkaloids. The biosynthesis gene cluster of pyridoxatin, *pdx*, was identified, while the gene clusters responsible for synthesizing trichodin A and asperpyridone A remain elusive. Interestingly, these compounds were found to accumulate in *topC*OE transformants, indicating that TopC has the capacity to simultaneously regulate the biosynthesis of these compounds. Further investigation through a literature search revealed that the gene clusters responsible for biosynthesis are mainly *tol*, *pdx*, and *lep*. Additionally, local blast analysis uncovered that within the *T. ophioglossoides* genome, the *topE* gene exhibits a remarkably high similarity to the core genes of these clusters, particularly *tolA*, with a similarity of up to 90%. Based on these findings, we postulate that the top gene cluster likely represents a multifunctional biosynthetic gene cluster responsible for producing pyridoxatin, trichodin A, and asperpyridone A.

### 3.3. Transcriptional Analysis of Top Gene Cluster

The *top* gene cluster, consisting of eight genes predicted by antiSMASH, was experimentally validated. To elucidate the gene expression dynamics within this cluster and delineate its boundaries, we conducted transcriptional analysis on the generated transformants using qRT-PCR (Appendix A). Our findings revealed a substantial increase in the transcription levels of *topA* through *topH* in the *topC*OE transformants when compared to the wild-type (WT) strain. Notably, other genes exhibited no significant alterations in their expression patterns (Figure 4). These results provide compelling evidence that the *top* gene cluster spans from *topA* to *topH*, encompassing eight genes across approximately 45 kilobases. Furthermore, this cluster can be selectively upregulated through the overexpression of the regulatory gene *topC*. 

### 3.4. TopC Positively Regulates Pyridoxatin, Trichodin A, and Asperpyridone A Biosynthesis by Binding All the Promoters of the Top Gene Cluster

The mechanism governing the accumulation of six compounds resulting from the overexpression of the *topC* gene has hitherto remained elusive. To address this knowledge gap, we conducted in vitro Electrophoretic Mobility Shift Assay (EMSA) experiments, aiming to decipher the intricate interplay between the TopC protein and the promoters of the aforementioned *top* structural genes.

GAL4, a well-known transcription factor, typically recognizes and binds to gene promoters through its N-terminal DNA-binding domain while activating transcription factors via its C-terminal activation domain. Therefore, we endeavored to selectively express the N-terminal DNA-binding domain (DBD) of TopC in *E. coli* BL21, facilitated by Trx soluble tag proteins. Subsequently, the purified fusion protein, with a molecular weight of 32 kDa, was subjected to the ensuing DNA binding experiments (Figure 5). We amplified five distinct DNA probe fragments from *T. ophioglossoides* genomic DNA, utilizing FM-labeled primers. These five DNA probes, along with a promoter of an un-specific gene selected from the *T. ophioglossoides* genome, were then subjected to interaction studies with TopC-DBD protein. The results of the EMSA experiments unequivocally demonstrated the binding affinity of TopC-DBD protein to all five probes in vitro, discernibly differentiating them from the control group featuring a promoter of an un-specific gene (Free vector probe) (Figure 5 and Appendix A). Remarkably, as the protein concentration increased, the probes exhibited a higher frequency of binding interactions and migrated shorter distances within the gel, ultimately forming discernable delayed probe-TopC-DBD complex bands. As depicted in Figure 5, it is apparent that the protein’s affinity for the *topB* promoter is notably lower compared to its binding affinity with the promoters of other genes within this cluster. Notably, when binding to promoters designated as C or A–H, a staircase emerged, characterized by a progressive increase in size, suggesting a looser binding of the TopC protein. Significantly, EMSA experiments involving C or A–H probes provided compelling evidence of their ability to bind to TopC-DBD.

Building upon these outcomes, we can postulate a regulatory mechanism for TopC. In this proposed mechanism, TopC directly engages with the upstream promoter regions of all genes encompassed within the top gene cluster, thereby augmenting the expression levels of the relevant genes.

Based on our experimental findings, we hypothesize the existence of a conserved binding site for the TopC protein within the promoter regions of all genes. To explore this further, we employed the MEME suite tool (https://meme-suite.org accessed on 20 April 2023) to identify potential DNA binding motifs within these promoter regions. Through MEME-ChIP analysis, we successfully pinpointed a potential TopC consensus binding site with the highest score, ATCGTTGTGTTTATTTGTTT, which was consistent across five promoter regions.

### 3.5. Identification of Putative Biosynthetic Pathway of Pyridoxatin, Trichodin A, and Asperpyridone A in T. ophioglossoides 

To validate our aforementioned hypothesis concerning this gene cluster, we employed homologous recombination to delete the key short-chain dehydrogenase gene, *topH* within the *topC*OE background strain (Appendix A). As depicted in the culture broth profile of the mutant *topC*OEΔ*topH* (Figure 6), compounds **3**, **4**, and **6** were no longer detectable, strongly suggesting their synthesis through the *top* gene cluster. This gene cluster, responsible for the production of trichodin A and asperpyridone A, represents a novel discovery in *T. ophioglossoides*. We delved into the biosynthetic pathways of this *top* biosynthetic gene cluster through gene knockout experiments, metabolite identification, and the study of homologous enzymes. A homologous gene cluster, *tol*, was identified via homology alignment. The core PKS-NRPS enzyme TopE/TolA, enoyl reductase TopD/TolC, cytochrome p450 TopF/TolD, and cytochrome p450 TopG/TolB, respectively, share 90.1%, 81%, 82.3%, and 96.8% sequence similarity at the amino acid level (Appendix A), indicative of their likely identical functions [15]. Furthermore, TopB, an annotated enzyme in this gene cluster, is predicted to be an o-methyltransferase (OMT) with a sequence homology of 99.6% similarity to AdxI, a protein previously implicated in pyridoxatin, trichodin A, and asperpyridone A biosynthesis [23]. Finally, SDR TopH is the putative ketoreductase, exhibiting an overall homology of 91.7% with PdxG, while TopA shares an 89.6% homology with PdxF within the *pdx* gene cluster, suggesting potentially similar functions.

In the culture broth profile of the *topC*OEΔ*topH* mutant, compounds **3**, **4**, **5**, and **6** were conspicuously absent, while two peaks, **7** and **12**, were detected, albeit in trace amounts. Due to the extremely limited yield of compound **12**, we encountered difficulties in obtaining a sufficient quantity for nuclear magnetic resonance (NMR) analysis. However, we successfully determined its molecular formula as C_15_H_21_NO_4_ via LC-MS/MS. The UV spectrum of compound **12** closely resembled that of tolypyridone D, a known compound in the literature, with the only difference being an additional hydroxyl group in compound **12** compared to tolypyridone D (**9**) [15]. Therefore, we postulate that compound **12** represents a derivative of tolypyridone D (Figure 6 and Figure 7, Appendix A), suggesting that TopH may possess a broader substrate tolerance compared to PdxG [23]. 

In the metabolite profile of *topC*OE*ΔtopA*, compounds **4**, **5**, and **6** were notably absent, and compound **9**, along with its derivatives, was not detected, implying a role for TopA in the biosynthesis of intermediate **9** (Appendix A). Further investigations are warranted to elucidate the precise catalytic mechanism of TopA.

In the knockout strains of *topC*OEΔ*topH* and *topC*OE*ΔtopA*, the production of compound **5** was undetectable. Compound **5** naturally exhibits a low yield in the *topC*OE strain. It is believed to be generated through a single hydroxylation step from compound **2**, as indicated by their molecular formulas. The minimal yield of compound **5** led us to speculate that this reaction might occur through a minor pathway with low enzyme expression or potentially be catalyzed by an enzyme external to the cluster. 

In summary, we propose a biosynthetic pathway for pyridoxatin, trichodin A, and asperpyridone A in *T. ophioglossoides*, as depicted in Figure 7, building upon previous biosynthesis studies [15,23].

## 4. Discussion

Filamentous fungi possess the remarkable capacity to produce a diverse array of structurally intricate secondary metabolites with multifaceted biological activities. Among these, pyridone alkaloids constitute a highly diverse and bioactive subgroup [16]. However, the majority of gene clusters responsible for pyridone alkaloid biosynthesis remain enigmatic. Notably, asperpyridone A, an uncommon pyridone alkaloid initially isolated from the endophytic fungus *Aspergillus* sp. TJ23, has exhibited a remarkable glucose uptake effect in liver HepG2 cells, surpassing metformin in efficacy [29]. Trichodin A, on the other hand, was derived from the marine fungus *Trichoderma* sp. MF106 and demonstrated antibiotic properties against the clinically significant microorganism, *Staphylococcus epidermidis* [28]. Yet, our understanding of the regulatory mechanisms governing their biosynthesis remains elusive, thereby impeding their potential clinical applications.

Various Zn(II)_2_Cys_6_-type zinc finger transcription factors (TFs) have been characterized across different filamentous fungi, including *M. pilosus* [35], *A. nidulans* [33], and *P. citrinum* [36], and are presumed to serve as positive regulators of secondary metabolite biosynthesis. Here, we highlight a latent *T. ophioglossoides* PKS-NRPS gene cluster harboring a putative regulator, TopC, possessing a GAL4-type Zn_2_Cys_6_ binuclear cluster DNA-binding domain. In our investigation, TopC emerges as a key positive regulator governing the biosynthesis of asperpyridone A and trichodin A. Overexpression of the regulatory gene *topC* homologously triggers the activation of the cryptic gene cluster *top*, resulting in the production of asperpyridone A and trichodin A. TopC exerts direct transcriptional activation of the structural genes within the *top* cluster by binding to their respective promoters. 

It is important to note that microbial metabolic regulation operates as a complex network, and the expression of the *topC* gene may be subject to control by other upstream regulatory proteins. In typical growth conditions, secondary metabolites are non-essential for microbial growth, and gene clusters involved in secondary metabolism often remain in a repressed or silenced state. In response to specific environmental cues or stressors, microbes initiate signaling pathways that activate upstream regulatory proteins. Subsequently, these upstream regulators further stimulate downstream proteins, including TopC. TopC, in turn, binds to the promoters of genes within the *top* cluster, recruits transcription-associated enzymes, and instigates gene transcription. This intricate process warrants further exploration. In our study, we have provided the initial characterization of the gene cluster responsible for asperpyridone A and trichodin A biosynthesis in fungi, shedding light on their biosynthetic pathway through an elucidation of the deduced gene functions within the *top* biosynthetic gene cluster.

## 5. Conclusions

In summary, our study unveiled a previously unrecognized biosynthetic gene cluster, named *top*, responsible for pyridone alkaloid production in *T. ophioglossoides* through comprehensive genome analysis. Notably, the overexpression of the pathway-specific regulatory gene, *topC*, residing within the *top* cluster, resulted in the activation of this cryptic gene cluster. Consequently, this led to the substantial accumulation of three significant pyridone alkaloids: asperpyridone A, trichodin A, and pyridoxatin. Our investigations unveiled TopC as a positive regulator governing the biosynthesis of pyridoxatin, trichodin A, and asperpyridone A. This regulatory protein achieves this by directly recognizing the promoter regions of all the *top* gene cluster’s constituent genes and binding to them. Furthermore, we made a pioneering discovery by identifying the asperpyridone A and trichodin A biosynthetic gene cluster in *T. ophioglossoides* through targeted key enzyme gene deletions. Our proposed biosynthesis mechanisms for these compounds emerged from a thorough analysis of gene deletions, metabolite profiles, and the functions of homologous enzymes. Notably, we identified an intriguing short-chain dehydrogenase, TopH, which exhibits remarkable flexibility in reducing various ketones, adding an additional layer of complexity to the biosynthetic pathway.

## Figures and Tables

**Figure 1 microorganisms-11-02578-f001:**
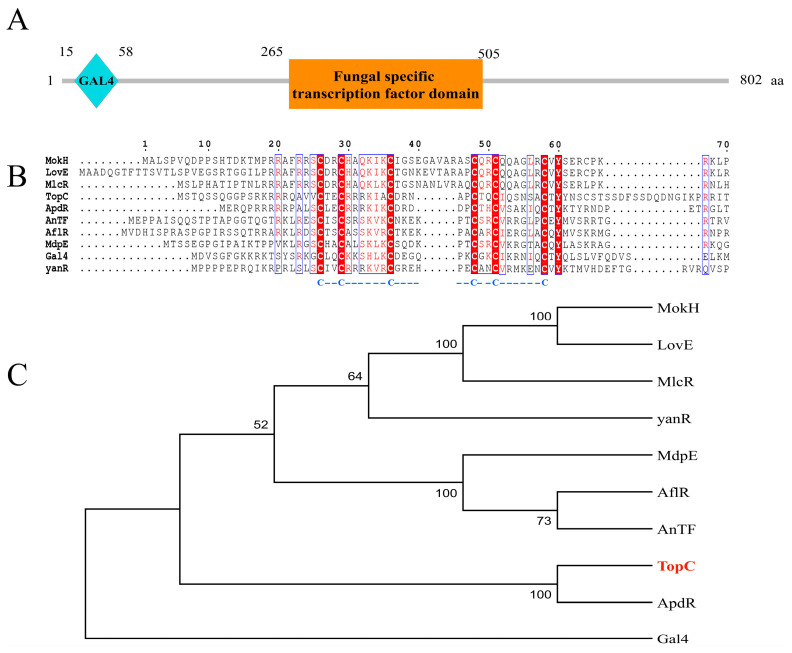
Characterization of transcription factor TopC and its phylogenic analysis. (**A**) domain characterization of transcription factor TopC in top cluster. (**B**) alignment analysis of the conserved cysteine amino acids. the conserved cysteine amino acids are marked in blue font below image B. (**C**) evolutionary phylogenetic analysis via MEGA7. *An*yanR (Accession no. G3Y415.1), GAL4 (Accession no. QGN14419.1), *Ap*AflR (Accession no. P43651.3), MdpE (Accession no. AN0148), *An*TF (Accession no. AAC49195), *Pc*MlcR (Accession no. Q8J0F2.1), *Mp*MokH (Accession no. Q3S2U4.1), *At*LovE (Accession no. Q0C8L8.1), *Cg*ApdR (Accession no. XP_045268485.1).

**Figure 2 microorganisms-11-02578-f002:**
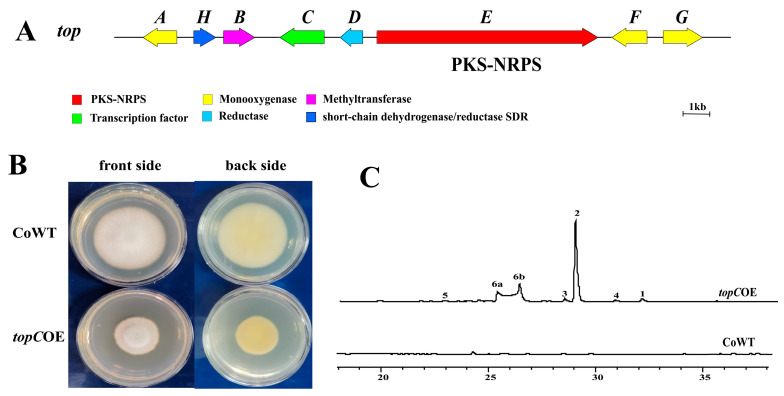
Schematic representation of the *top* gene cluster and morphological characteristics, as well as HPLC chromatographs of transformants. (**A**) the schematic representation illustrates the *top* gene cluster, encompassing eight genes from *T. ophioglossoides*. This cluster encompasses a PKS-NRPS hybrid enzyme (*topE*) and various modified enzymes. These enzymes include one hypothetical methyltransferase (*topB*), a short-chain dehydro-genases/reductases (*topH*), three cytochrome p450 monooxygenases (*topA*, *F, G*), one enoyl reductase (*topD*), and one C6 transcription factor (*topC*) (**B**) a total of 10^3^ spores were inoculated on a PDA plate and incubated at 26 °C for 7 days. The *topC*OE mutants displayed a deeper yellow color compared to Co-WT on the reverse side (**C**) HPLC profiles of the culture broth from topCOE and Co-WT were obtained at a wavelength of 280 nm. topCOE exhibited six distinct compound peaks in contrast to Co-WT.

**Figure 3 microorganisms-11-02578-f003:**
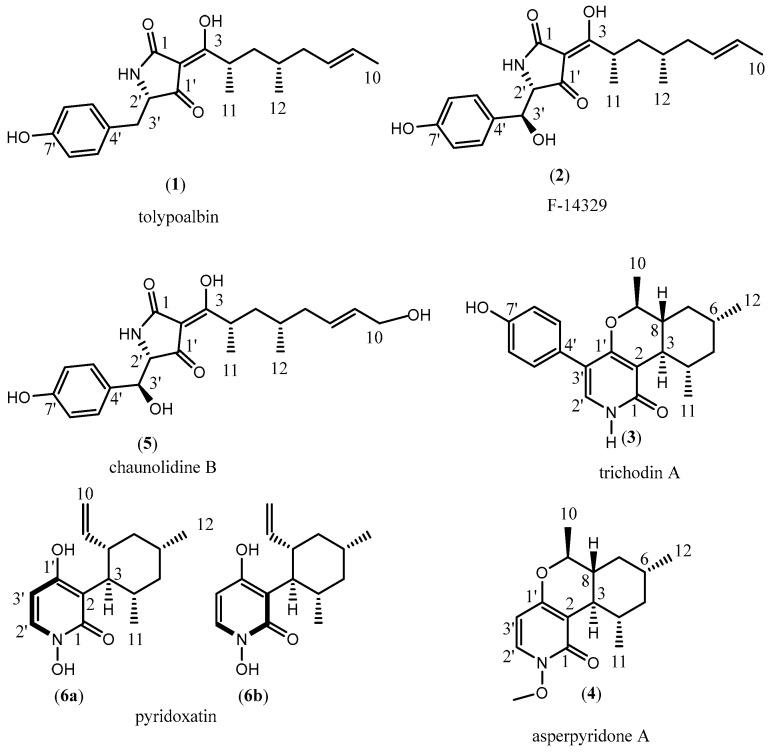
Structural determination and the chirality of compound **1**–**6** produced by *topC*OE.

**Figure 4 microorganisms-11-02578-f004:**
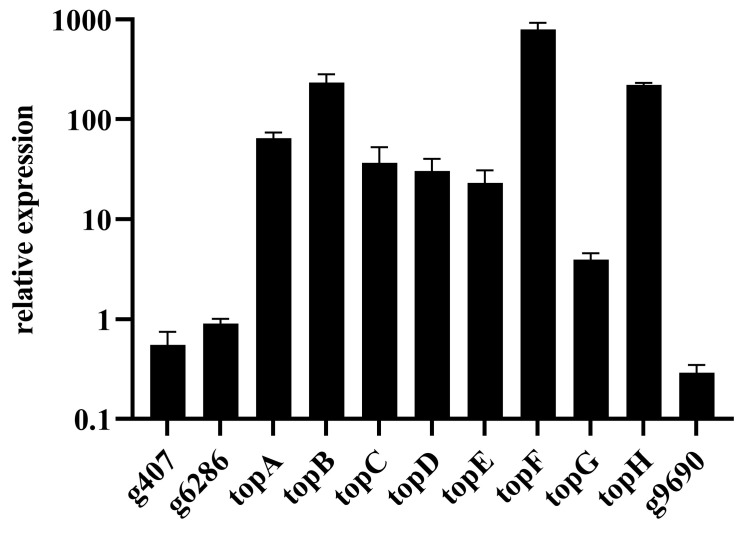
Determination of the borders of the *top* cluster via qRT-PCR in *topC*OE. The translation elongation factor *tef* was used as the internal controls.

**Figure 5 microorganisms-11-02578-f005:**
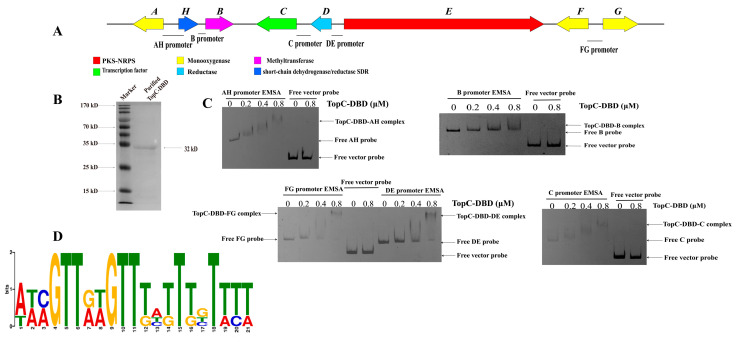
Heterologous expression of TopC and analysis of its affinity binding to promoter DNA via EMSA. (**A**) we designed the promoters within the *top* gene cluster. (**B**) The expression of TopC-DBD with a His-tag in *E. coli* was achieved successfully, as depicted by the rightward arrow, yielding purified topC-DBD protein with a molecular weight of 32 kD. (**C**) Affinity binding analysis of TopC to five gene promoters within *top* via EMSA. (**D**) the conserved binding motif of TopC was predicted via MEME analysis.

**Figure 6 microorganisms-11-02578-f006:**
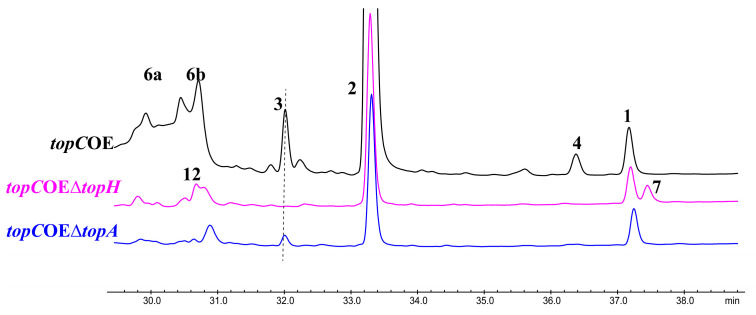
HPLC profiles of the culture broth of *topC*OE, *topC*OE*ΔtopA,* and *topC*OEΔ*topH* (λ = 280 nm).

**Figure 7 microorganisms-11-02578-f007:**
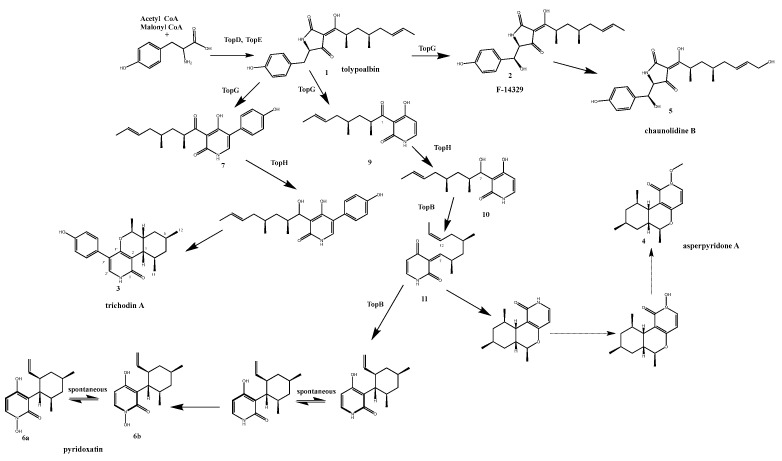
Biosynthetic Pathways of pyridoxatin, trichodin A, and asperpyridone A in *T. ophioglossoides*.

## Data Availability

The data presented in this study are available upon request from the corresponding author. The RNA-seq data are not publicly available because other data from these whole-genome transcriptomes are being used for other analyses to be published independently of this one.

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
