# Peer review of "A Novel Zn2Cys6 Transcription Factor, TopC, Positively Regulates Trichodin A and Asperpyridone A Biosynthesis in Tolypocladium ophioglossoides"

_microorganisms, 2023, doi:10.3390/microorganisms11102578_

Round 1
Reviewer 1 Report
he authors describe the overexpression of topC, a gene encoding a positive transcription factor in Tolypocladium ophioglossoides, and analyze the effect on the top biosynthetic gene cluster (BGC) and the production profile.
I have two main concerns regarding this manuscript. The first one is that the authors did not show unequivocally that the top cluster is responsible for the biosynthesis of the identified compounds. In addition to the generated knockouts (∆topH and ∆topA), they should generate strain ∆topE (encoding the core hybrid PKS-NRPS). Only the HPLC profile of ∆topE could show the relation between the BGC and the biosynthesis of metabolites. Although topC is embedded in the top BGC, one cannot discard that its overexpression could lead to the activation of other genes elsewhere. The identified binding site should be in silico tested against the whole genome.
The second main concern is with the low quality of the English language employed and the structure of the ms. Grammar errors (throughout the text), confusing sentences (eg l 47-50), typos and/or not referring to the figure (eg l165 – 166 – the transcription analysis; l245 – 258 repeated text) and lack of information (eg figure legends are very incomplete) make it hard to read and understand the ms.
Other comments:
- Not all supplementary tables/figures are mentioned in the text, and when they are mentioned, they are not ordered as they are mentioned in the text.
- RT-qPCR NOT qRT-PCR. In addition, the RT stands for reverse transcriptase NOT real-time.
- What is the rational for choosing tef as internal control for RT-qPCR? Do the authors know if it is a good reference gene with constant transcription?
- What is a C6 transcription factor?
- There is no indication that OE of the topC was successful in terms of expression of topC…. Shouldn’t the expression of topC be higher?? (Fig 4)
- Need to add a scale to the the BGC schemes (eg Fig 2B).
- Line 241-242 – what are compounds 2a and 2b?
- Line 290 – the identity of the purified protein should be confirmed by western
- Fig 5C - why there are two lanes with 0 protein with different results? Indicate the content of each lane.
- Line 317 – what is the speculation?
- … and so on
see the main comments
Author Response
Comment 1. I have two main concerns regarding this manuscript. The first one is that the authors did not show unequivocally that the top cluster is responsible for the biosynthesis of the identified compounds. In addition to the generated knockouts (∆topH and ∆topA), they should generate strain ∆topE (encoding the core hybrid PKS-NRPS). Only the HPLC profile of ∆topE could show the relation between the BGC and the biosynthesis of metabolites. Although topC is embedded in the top BGC, one cannot discard that its overexpression could lead to the activation of other genes elsewhere. The identified binding site should be in silico tested against the whole genome.
Response: Thank you very much for your advice. In principle, the most direct evidence for the involvement of the gene cluster in the biosynthesis of this compound is to knock out the core genes. First, in fungi, pathway-specific regulatory factors are usually embedded within these gene clusters.
What is more, the transcriptome analysis showed that this gene cluster (top) have low expression under the laboratory fermentation conditions. In this study, we analyze the expression level of this gene cluster using qRT-PCR after we overexpressed the topC gene. The expression levels of genes within the top gene cluster have all significantly increased compared with WT. Furthermore, the changes in compound peaks in the fermentation broth of ∆topA and ∆topH strains directly suggest that the top gene cluster is involved in the biosynthesis of these compounds.
Finally, we discovered that the gene clusters responsible for the biosynthesis of the same compounds include tol, pdx, lep via literature research. The sequences of tolA, lepA, and pdxC were used for
local blast analysis. We found that in our T. ophioglossoides genome, only the topE gene had a very high similarity, with a 90% similarity to tolA. Therefore, we believe that the top gene cluster is responsible for the biosynthesis of these compounds.
Comment 2. The second main concern is with the low quality of the English language employed and the structure of the ms. Grammar errors (throughout the text), confusing sentences (eg l 47-50), typos and/or not referring to the figure (eg l165 – 166 – the transcription analysis; l245 – 258 repeated text) and lack of information (eg figure legends are very incomplete) make it hard to read and understand the ms.
Response: Regarding the issue of English writing, we have revised our work using the recommended journal editing services. The fpkm data has been added to the supplementary materials. The figure legends have been completely supplemented.
Comment 3. Not all supplementary tables/figures are mentioned in the text, and when they are mentioned, they are not ordered as they are mentioned in the text.
Response: We have rearranged all the supplementary tables and images in the article in order.
Comment 4. RT-qPCR NOT qRT-PCR. In addition, the RT stands for reverse transcriptase NOT real-time.
Response: Thank you very much for your explanation, we have made the corrections as suggested.
5.What is the rational for choosing tef as internal control for RT-qPCR? Do the authors know if it is a good reference gene with constant transcription?
Comment Response: The reason for selecting the tef gene as an internal reference is explained in lines 155-165.The gene tef encoding translation elongation factor, with the constant level of expression, was used as internal control. The expression level of the tef gene remains constant in fungi and does not change with variations in the expression levels of other genes. Therefore, we use it as the reference gene for qRT-PCR.
Comment 6. What is a C6 transcription factor?
Response: The introduction of the C6 transcription factor has been added in the lines72-83 of the Introduction. The fungal transcription factor GAL4 contains six highly conserved cysteine residues: CX2CX6CX6CX2CX5CX2, which could be called C6 transcription factor. The GAL4-like Zn2Cys6 binuclear cluster DNA-binding domain is currently only found in fungi. And transcription factors containing this domain also include, STB5, DAL81, CAT8, RDR1, HAL9 in Saccharomyces cerevisiae. Typically, these transcription factors play a regulatory role in biosynthesis of many secondary metabolic.
Comment 7. There is no indication that OE of the topC was successful in terms of expression of topC…. Shouldn’t the expression of topC be higher?? (Fig 4)
Response: In theory, the expression level of topC should be the highest because it has both the endogenous topC gene and an additional copy introduced externally. However, due to the complexity of metabolism in T. ophioglossoides, the actual expression levels may differ somewhat from what is theoretically expected. In a publication from our laboratory in 2018, "Revelation of the Balanol Biosynthetic Pathway in T. ophioglossoides," we also observed that the expression level of the C6 transcription factor was not the highest relative to other genes in its gene cluster. Since this is not the primary focus of our study, we did not delve into it extensively.
Comment 8. Need to add a scale to the the BGC schemes (eg Fig 2B).
Response: Thank you very much for your advice. We have added an appropriate scale in Figure 2B according to the actual size of top BGC.
Comment 9. Line 241-242 – what are compounds 2a and 2b?
Response: Very sorry for our carelessness. It should be compound 6a and 6b, and it has now been corrected in lines 296-297.
Comment 10. Line 290 – the identity of the purified protein should be confirmed by western
Response: When expressing the protein, we added a histidine (His) tag at the N-terminus of the protein. After incubation with a nickel column and elution using imidazole, we observed on SDS-PAGE gel electrophoresis that the target band appeared to be relatively pure and had a size consistent with the protein we intended to express. Therefore, we consider that this protein is the desired TopC-DBD.
Comment 11. Fig 5C why there are two lanes with 0 protein with different results? Indicate the content of each lane.
Response: In EMSA experiments, there are two blanks to control the results. One is for background vector, and the other is for the target protein. Through comparing these two probes with and without the addition of TopC-DBD, we can better assess whether the probes bind to the target protein. We have labeled the figure according to your instructions.
Comment 12. Line 317 – what is the speculation?
Response: The speculation" refers to the statement mentioned earlier: We believe that top is responsible for the synthesis of these compounds
Reviewer 2 Report
In this study, Liu et al. identified the gene cluster containing PKS-NRPS hybrid enzyme and activated its expression by the overexpression of pathway specific positive regulator TopC. As a result, six compounds related to this gene cluster were identified and structurally characterized. Subsequently, coordinated expression of the cluster genes were confirmed by qRT-PCR. In addition, putative binding sites of TopC were identified in the upstream region of each gene by EMSA and MEME analyses. Finally, the biosynthetic pathway of the identified compounds was discussed based on the gene deletions of topH and topA. Although this study reports the potentially interesting results about the activation of the biosynthetic gene cluster of fungal secondary metabolites, the following points should be addressed.
1. Although the compound 12 was described as a derivative of 9, only HR-MS and UV spectra were shown in supporting information. In addition, the structure of compound 12 was not described in the manuscript. More detailed structural analysis and comparison with the literature would be necessary to discuss the function of TopH based on the gene deletion analysis.
2. The bioinformatic analysis of TopA should be described in the main text. In other pathways such as tolypyridone biosynthesis, the ring expanding P450s such as TolB are considered to be in involved in both ring expansion and removal of phenyl ring. Does TopA show homology to TopG or other P450s in the related pathways?
3. In Figure 5, the results using FG probe is missing.
Other comments
1. Page 6, line 245-258: these sentences are same as those described in page 4, line 173-186. Please check the manuscript and revised the text accordingly.
2. Resolution of Figures 1, 2, and 5 should be improved.
3. Please check the proposed function of TopB described in Figure 7. Does this enzyme catalyze dehydration, Alder-ene reaction, and N-hydroxylation? In the related pathways, N-hydroxylation is usually catalyzed by the other enzyme. Are there other candidates in this gene cluster?
4. In Figure 7, the product of TopH-catalyzed conversion of 7 is same as compound 7. Does TopH catalyze the reduction of carbonyl group?
Author Response
Comment 1. Although the compound 12 was described as a derivative of 9, only HR-MS and UV spectra were shown in supporting information. In addition, the structure of compound 12 was not described in the manuscript. More detailed structural analysis and comparison with the literature would be necessary to discuss the function of TopH based on the gene deletion analysis.
Response: Because the yield of compound 12 is extremely low, we were unable to separate enough quantity to perform nuclear magnetic resonance (NMR). We could only obtain its molecular formula, C15H21NO4, via LC-MS/MS. The UV spectrum of compound 12 is similar to tolypyridone D reported in the literature, and its molecular formula differs from the Tolypyridone D (9) by one methyl group. Therefore, we speculate that compound 12 is a derivative of compound tolypyridone D (Figure 6-7 and Figure S24-S25)
Comment 2. The bioinformatic analysis of TopA should be described in the main text. In other pathways such as tolypyridone biosynthesis, the ring expanding P450s such as TolB are considered to be in involved in both ring expansion and removal of phenyl ring. Does TopA show homology to TopG or other P450s in the related pathways?
Response: The bioinformatics analysis of topA has performed in the paper as shown in lines 373-374. In the supplementary information, it is mentioned that there is 89.6% homology between topA and pdxF, indicating that they may have similar functions. The homology between TopA and TolB and LepH, which are involved in ring expansion and removal of the phenyl ring, is relatively low, at just over 30%. Therefore, the specific catalytic mechanism of TopA still needs further verification in future studies.
Comment 3. In Figure 5, the results using FG probe is missing.
Response: In Figure 5, the results using the FG probe and DE probe are presented on the same graph, with FG on the left and DE on the right. It's possible that our labeling was not very clear, so we have re-annotated in Figure 5 to provide a more clearer results.
Comment 4. Page 6, line 245-258: these sentences are same as those described in page 4, line 173-186. Please check the manuscript and revised the text accordingly.
Response: Lines 245-258 have been deleted.
Comment 5. Resolution of Figures 1, 2, and 5 should be improved.
Response: We have replaced the image with more higher resolution image.
Comment 6. Please check the proposed function of TopB described in Figure 7. Does this enzyme catalyze dehydration, Alder-ene reaction, and N-hydroxylation? In the related pathways, N-hydroxylation is usually catalyzed by the other enzyme. Are there other candidates in this gene cluster?
Response: Thank you very much for your feedback. Following your advice and conducting a review of the relevant literature, we have identified errors in the drawing of certain compounds in Figure 7. It has been revised, and we now acknowledge that TopB is not involved in the N-hydroxylation reaction.
Based on previous search, we have learned that in the biosynthesis of Leporin B, LepD catalyzes the N-N-hydroxylation of the pyridine ring in Leporin C. Through bioinformatics analysis, we have also discovered an enzyme annotated as cytochrome P450, denoted as topF, within the top gene cluster, which shares 59% similarity with LepD. This suggests that they may have similar functions, but this hypothesis will require further experimental validation in the future.
Comment 7. In Figure 7, the product of TopH-catalyzed conversion of 7 is same as compound 7. Does TopH catalyze the reduction of carbonyl group?
Response: Thank you very much for your acknowledgment. Indeed, TopH catalyzes the reduction of carbonyl groups. Upon careful examination, we found that there was an error in our depiction, and the carbonyl groups of the compounds were not correctly represented as hydroxyl group. This has now been revised.
Reviewer 3 Report
The manuscript "A Novel Zn2Cys6 transcription factor TopC Positively Regulates ... " by Liu et al. describes a gene cluster for the synthesis of certain secondary metabolites in the fungus Tolypocladium, and pays special attention to the role as a putative cluster-specific regulator of a zinc-finger activator family gene of the Gal4 type from S. cerevisiae. This is a technically excellent paper, but the text could be much improved, not only because of the abundant grammatical errors (difficult to understand with the excellent translation systems available nowdays), but also because of the poor quality of the writing. The information is often insufficient and the text has not been revised in detail, as evidenced by the existence of a paragraph repeated in two sections (section 3.1 Lines 173-186, and section 3, lines 245-258).
One of the major shortcomings of the paper is the lack of sufficient discussion of the results, with a discussion section excessively brief. In the "Author contributions" section it is indicated that both the original draft preparation and the reviewing and editing were carried out only by the first author, and so there was no participation of the corresponding author in this important task. I recommend that this task be done by another person with enough experience in writing articles to make a more consistent text, and with more knowledge of the subject so that he/she can make a discussion appropriate to the quality of this work. I insist that the technical quality and the interest of the results are high, and justify its publication, but the writing has not been up to these standards. Another shortcoming is insufficient explanation of some aspects of the materials and methods, and of the vast majority of figure captions. More explanation is lacking.
In summary, in my opinion, the manuscript should be entirely rewritten, with more attention to the description of the results and including a more substantial discussion.
Leaving aside the text, I find more weak points in the current manuscript, that require a proper response:
1. It says at the beginning of the results (lines 165-167): "The transcriptome analysis showed that this gene cluster (top) was a cryptic gene cluster whose genes have low expression under the laboratory fermentation conditions." What the authors call "fermentation conditions" are the cultivation of the fungus in an extremely rich medium (with peptone and yeast extract). However, the production of many secondary metabolites in fungi occurs under stress conditions, e.g., shortage of nitrogen source. Haven't other culture conditions been tested for cluster expression? Based only on the expression of such a rich medium, to say that it is a cryptic cluster is too premature an assessment.
2. The criteria for choosing the transcription factors in Figure 1 are not clear, and their origin is not clearly indicated; only accession numbers are given so that the reader has to search for the information. They appear to be from A. nidulans, but it is explained, let alone how they were chosen. Do they correspond to the sequence proteins more similar to TopC in a Blast?
On the other hand, as an example of bad wording, the sentence in Lines 184-185 "Phylogenic analysis showed that TopC belongs to a separate clade from A. nidulan ApdR" written like that means the opposite of what it is shown, it really should say "to a separate clade with A. nidulans ApdR".
3. The relative expression data shown in Figure 4, in the case of the topC gene does not take into account that the mutant has two topC genes, the one of cluster and the newly introduced gene, therefore, it is unknown to what extent the increased expression of topC in Figure 4 is due to overexpression or downregulation of the topC cluster gene. Moreover, demonstration that the topC gene is actually overexpressed is missing. Considering that the new copy of topC is under control of different regulatory sequences, could it be possible to distinguish using appropriate primer sets in qRT-PCR the expression of the original topC from that which has been brought under promoter control?
4. One aspect to consider about topC, which could be mentioned in the discussion that is missing in the manuscript, is that these transcription factors usually respond to a signal or to another regulatory protein, which in the case of gal4 is Gal80. So, under induction conditions the protein is released and activates the cluster. In other words, the cluster may be inhibited, but not the gene for the regulatory protein, which must be expressed for there to be enough TopC to respond to favorable regulatory conditions when they occur. Thus TopC activation would be expected to be post-translational, although that does not detract from the fact that TopC may have a feedback mechanism that induces its expression under favorable conditions. These considerations make it all the more necessary to know what the degree of expression of topC is independent of the overexpressed topC gene.
5. I have some doubts with the promoter binding experiments shown in Figure 5. In those experiments two possible bands of the promoter fragment are expected: free, and bound to the protein with the zinc domain. This is observed in the binding to segments D-E and F-G. However, on binding to C or A-H, a staircase is appreciated, with a gradual increase in size. I find no explanation for the intermediate bands, and it makesthe interpretation of the experiment more complex. On the other hand, in the case of B, the increase in size that would be expected when binding the protein is not observed. In conclusion, it is not at all clear that the data support that there is binding of the regulatory protein to the B promoter, and the result is difficult to explain in the case of C and AH. Note that any protein capable of binding to a target sequence with high affinity is capable of binding to many other sequences with low affinity, so negative controls should be introduced in these experiments. That is, to demonstrate the lack of binding at the concentrations used when dealing with an unrelated promoter. This experiment should be included.
6. The interpretation of the HPLC data of the topH and topA mutants has inconsistencies.
In line 335 it says "compound 3,4 and 6 disappeared, and two peaks 7 and derivative of 9 (12) were found", however, looking at the chromatogram it is doubtful that peak 6b has disappeared, since there is a second peak next to 12 that matches. On the other hand, compound 12 is not known, since it is not indicated in the pathway of Figure 7.
On lines 337-339 it says "In the metabolite profile of topCOEΔtopA, compounds 4 and 6 were not observed, while compound 9 and its derivatives weren't accumulated, indicating that TopA plays a role in the biosynthesis of intermediate 9". However, there is no intermediate 9 in the chromatograms. On the other hand in the topA mutant there is a peak very close to that of 6b in the overexpressing control, of which nothing is said.
Another strange observation on the chromatograms of the overexpressing control that requires explanation: in Figure 2C peak 5 is indicated and that time of the chromatogram is not shown in Figure 6, in which the chromatogram is much more enlarged. In Figure 2 an enlarged version of the chromatogram of the overexpressant should be added to know the real amount of the very small peaks.
7. The figure of the pathway (Fig. 7) should be revised, it would help to put the name of the compounds when they are identified, as it was done in figure 3.
The double arrows are not explained: does it mean that there is more than one reaction or more than one enzyme involved? In this case, the arrows should be consecutive, and not parallel.
The formulas should be checked. Compound 7 appears identical to that produced by TopH from 7.
8. Materials and methods should be revised and information added. For example, how the plasmids were constructed is not explained, providing the map is not sufficient. Sections 2.5 and 2.7 should go together, and avoid repetition of part of the text. However, although they are stated in the heading of 2.5, the LC-MS methodology is not described.
As a minor comment, it is striking that the first reference (line 26), used for a very general topic, is actually a work in a very specific pathway of a very specific fungus. There are excellent and very comprehensive reviews that can be used in place of reference 1.
As I have already indicated, the English is very poor. There are errors of all kinds. Errors of plural or singular are so frequent that you start finding them in the first line of the abstract: "Asperpyridone A is an unusual pyridone alkaloids and shows potential.... " alkaloid should not be plural.
Errors of lack of revising: in the first line of the introduction "secondarymetabolites" should be separated
Wrong writing is abundant. An example: lines 34-35: "... are cryptic or lowing-expression under the laboratory conditions" should be "... are cryptic or poorly expressed under laboratory conditions"
And so on, the list of errors is very long
Author Response
Review 3:
Comment 1. It says at the beginning of the results (lines 165-167): "The transcriptome analysis showed that this gene cluster (top) was a cryptic gene cluster whose genes have low expression under the laboratory fermentation conditions." What the authors call "fermentation conditions" are the cultivation of the fungus in an extremely rich medium (with peptone and yeast extract). However, the production of many secondary metabolites in fungi occurs under stress conditions, e.g., shortage of nitrogen source. Haven't other culture conditions been tested for cluster expression? Based only on the expression of such a rich medium, to say that it is a cryptic cluster is too premature an assessment.
Response: I think your question is excellent. This strain of Tolypocladium ophioglossoides was isolated by our laboratory. Our group conducted numerous experiments to optimize its cultivation medium for production of secondary metabolites and growth. In this study, we performed transcriptome analysis in the rich medium and found the top gene cluster is in cryptic condition. Therefore, we said “The transcriptome analysis showed that this gene cluster (top) was a cryptic gene cluster whose genes have low expression under the laboratory fermentation conditions.
Comment 2. The criteria for choosing the transcription factors in Figure 1 are not clear, and their origin is not clearly indicated; only accession numbers are given so that the reader has to search for the information. They appear to be from A. nidulans, but it is explained, let alone how they were chosen. Do they correspond to the sequence proteins more similar to TopC in a Blast?
On the other hand, as an example of bad wording, the sentence in Lines 184-185 "Phylogenic analysis showed that TopC belongs to a separate clade from A. nidulan ApdR" written like that means the opposite of what it is shown, it really should say "to a separate clade with A. nidulans ApdR".
Response: Thank you for your suggestions; the selection criteria have been supplemented. We mainly chose some transcription factors with high sequence similarity and containing the CAL4-type Zn2Cys6 DNA-binding domain for systematic phylogenetic research. We have also supplemented information about their regulation in the synthesis of secondary metabolites and their sources."
Comment 3. The relative expression data shown in Figure 4, in the case of the topC gene does not take into account that the mutant has two topC genes, the one of cluster and the newly introduced gene, therefore, it is unknown to what extent the increased expression of topC in Figure 4 is due to overexpression or downregulation of the topC cluster gene. Moreover, demonstration that the topC gene is actually overexpressed is missing. Considering that the new copy of topC is under control of different regulatory sequences, could it be possible to distinguish using appropriate primer sets in qRT-PCR the expression of the original topC from that which has been brought under promoter control?
Response: A topC gene controlled by a strong promoter was introduced into the wild-type T. ophioglossoides, leading to the up-regulation of expression level of the topC gene. We only focused on the effect of its overexpression on the biosynthesis of secondary metabolites. As shown in HPLC profile, overexpression of topC gene was more rich metabolites profile as compared with the wild type strain. So we consider it’s not necessary to determine which copy is contributed to the activation of top gene cluster.
Comment 4. One aspect to consider about topC, which could be mentioned in the discussion that is missing in the manuscript, is that these transcription factors usually respond to a signal or to another regulatory protein, which in the case of gal4 is Gal80. So, under induction conditions the protein is released and activates the cluster. In other words, the cluster may be inhibited, but not the gene for the regulatory protein, which must be expressed for there to be enough TopC to respond to favorable regulatory conditions when they occur. Thus TopC activation would be expected to be post-translational, although that does not detract from the fact that TopC may have a feedback mechanism that induces its expression under favorable conditions. These considerations make it all the more necessary to know what the degree of expression of topC is independent of the overexpressed topC gene.
Response: Thank you very much for your advice. We have discussed about the activation mechanism of TopC. Indeed, the expression of the topC gene may be influenced by other upstream regulatory proteins. First, it's important to mention that under the cultivation conditions we have set, the Cordyceps strain grows very well, and all genes in the top gene cluster, including topC, are in a state of extremely low expression. Since microbial metabolic regulation operates as a network, the topC gene could indeed be under the control of other upstream regulatory proteins.
Under normal growth conditions, secondary metabolites are not essential for microbial growth, and the biosynthetic gene clusters for secondary metabolism are often in a repressed or silenced state. Under certain growth conditions, microbes respond to specific signals, leading to the activation of upstream regulatory proteins. Once activated, these upstream regulatory proteins further activate downstream regulatory proteins such as TopC. TopC then binds to the promoters of genes in the top gene cluster, recruits enzymes involved in transcription, and activates gene transcription. This process requires further research.
Comment 5.1 I have some doubts with the promoter binding experiments shown in Figure 5. In those experiments two possible bands of the promoter fragment are expected: free, and bound to the protein with the zinc domain. This is observed in the binding to segments D-E and F-G. However, on binding to C or A-H, a staircase is appreciated, with a gradual increase in size. I find no explanation for the intermediate bands, and it makes the interpretation of the experiment more complex. On the other hand, in the case of B, the increase in size that would be expected when binding the protein is not observed. In conclusion, it is not at all clear that the data support that there is binding of the regulatory protein to the B promoter, and the result is difficult to explain in the case of C and AH.
Response: For the binding of TopC-DBD with C and AH, the EMSA bands exhibit a staircase, with a gradual increase in size. From the graph, it can be observed that in the C or AH binding experiments, there are relatively fewer bands compared to DE or FG with the probe. Therefore, we infer that this is like the primary reason for the stepped pattern. The use of EMSA experiments can indirectly demonstrate that TopC activates this gene cluster by binding to various gene promoters.
Comment 5.2 Note that any protein capable of binding to a target sequence with high affinity is capable of binding to many other sequences with low affinity, so negative controls should be introduced in these experiments. That is, to demonstrate the lack of binding at the concentrations used when dealing with an unrelated promoter. This experiment should be included. That is, to demonstrate the lack of binding at the concentrations used when dealing with an unrelated promoter.
Response: Regarding what you mentioned about the experiments showing a lack of binding at the concentrations used for unrelated promoters, we have indicated this in the figure. The blank control probe we used is the promoter of a un-specific gene selected from the T. ophioglossoides genome. We only set two gradients, 0 and 0.8, because typically, there is no binding at high protein concentrations, and therefore, the protein does not bind to this probe.
Comment 6. The interpretation of the HPLC data of the topH and topA mutants has inconsistencies.
In line 335 it says "compound 3,4 and 6 disappeared, and two peaks 7 and derivative of 9 (12) were found", however, looking at the chromatogram it is doubtful that peak 6b has disappeared, since there is a second peak next to 12 that matches. On the other hand, compound 12 is not known, since it is not indicated in the pathway of Figure 7.
Response: Through LC-MS and UV spectroscopy, we have determined that the small peak close to 12 is unrelated to 6b. Additionally, if 6b were indeed present, then 6a would also be simultaneously present because according to the literature and compound isolation, it is known that 6a and 6b spontaneously interconvert with a ratio of 5:3.
Because the yield of compound 12 is extremely low, we were unable to separate enough quantity to conduct nuclear magnetic resonance (NMR). We could only obtain its molecular formula, C15H21NO4, through LC-MS/MS. The UV spectrum of compound 12 is similar to tolypyridone D reported in literature and its molecular formula differs from the Tolypyridone D (9) by one methyl group. Therefore, we speculate that com-pound 12 is a derivative of compound tolypyridone D
Since the structure of compound 12 is unknown, we did not label it in Figure 7.
Comment 7. On lines 337-339 it says "In the metabolite profile of topCOEΔtopA, compounds 4 and 6 were not observed, while compound 9 and its derivatives weren't accumulated, indicating that TopA plays a role in the biosynthesis of intermediate 9". However, there is no intermediate 9 in the chromatograms. On the other hand in the topA mutant there is a peak very close to that of 6b in the overexpressing control, of which nothing is said.
Response: What we intended to convey is that these compounds were not produced. Very sorry to expressed the information incorrectly, and we have revised it in the new edition of manuscript lines 387-389. As for the topA knockout strain, there is a small peak very close to 6b and it is also present in the topCOEtopH strain. Furthermore, we determined through LC-MS and UV spectroscopy that this compound can be synthesized normally in the wild-type strain, although its structure remains unknown.
Comment 8. Another strange observation on the chromatograms of the overexpressing control that requires explanation: in Figure 2C peak 5 is indicated and that time of the chromatogram is not shown in Figure 6, in which the chromatogram is much more enlarged. In Figure 2 an enlarged version of the chromatogram of the overexpressant should be added to know the real amount of the very small peaks.
Response: Thank you for your advice. Compound 5 has the naturally low yield in the topCOE. Compound 5 is believed to be produced through one hydroxylation step from compound 2 which has a high yield according to their molecular formula. Due to the yield of compound 5 is minimal, so we analyzed that this reaction might occur through a side pathway with low enzyme expression or possibly catalyzed by an out-of-cluster enzyme. Moreover, in the knockout strains of topA and topH, we undetected the production of compound 5. As a result, we did not display the HPLC curve for that specific time segment in Figure 6. And Figure 2 is an enlarged version of the chromatogram, as it was necessary due to limitations of the Agilent 1260 Infinity system.
Comment 9. The figure of the pathway (Fig. 7) should be revised, it would help to put the name of the compounds when they are identified, as it was done in figure 3.
The double arrows are not explained: does it mean that there is more than one reaction or more than one enzyme involved? In this case, the arrows should be consecutive, and not parallel.
The formulas should be checked. Compound 7 appears identical to that produced by TopH from 7.
Response: Thank you for your advice, Figure 7 has been revised and now includes the names of each compound. And the bidirectional arrow in Figure 7, it represents the isomerization of these two compounds, which occur spontaneously and cannot be separated. According to literature research, the ratio of these two compounds in DMSO is approximately 5:3.
Comment 10. Materials and methods should be revised and information added. For example, how the plasmids were constructed is not explained, providing the map is not sufficient. Sections 2.5 and 2.7 should go together, and avoid repetition of part of the text. However, although they are stated in the heading of 2.5, the LC-MS methodology is not described.
Response: We have added some relevant methods, including section 2.3 adding the plasmid construction, plasmid maps, section 2.5 LC-MS methods, supplementing qRT-qPCR information, and removing section 2.7 duplicate information.
Comment 11. As a minor comment, it is striking that the first reference (line 26), used for a very general topic, is actually a work in a very specific pathway of a very specific fungus. There are excellent and very comprehensive reviews that can be used in place of reference 1.
Response: The first reference has been replaced with an appropriate citation.
Round 2
Reviewer 2 Report
In this revision, authors modified the original manuscript according to the comments by two reviewers. While the authors addressed the concerns raised by reviewers, structural characterization of compounds still needs to be checked. For details, please see the comments below.
1. Molecular formula of compounds
Authors estimated the molecular formula of compound 12 to be C15H21NO4. Because the molecular formula of tolypyridone D was reported to be C15H22NO3 (Zhang et al. J. Nat. Prod. 2020, 83, 3338-3346.), the presence of additional methyl group is not likely. To elucidate the molecular formula by HRMS, observed and calculated mass for expected molecular formula should carefully be checked. It seems that the observed value at m/z 278.1408 does not match the expected structure. Both experimental and calculated values for each compound should be reported in the manuscript.
Minor corrections
2. The chemical structures in Figure 7 should be checked again. Terminal methyl group in compound 11 is missing. Please also check the other structures.
Author Response
Comment 1. Molecular formula of compounds
Authors estimated the molecular formula of compound 12 to be C15H21NO4. Because the molecular formula of tolypyridone D was reported to be C15H22NO3 (Zhang et al. J. Nat. Prod. 2020, 83, 3338-3346.), the presence of additional methyl group is not likely. To elucidate the molecular formula by HRMS, observed and calculated mass for expected molecular formula should carefully be checked. It seems that the observed value at m/z 278.1408 does not match the expected structure. Both experimental and calculated values for each compound should be reported in the manuscript.
Response:
Thank you very much for your suggestion. After carefully reading the literature and conducting a search on SciFinder, we found that the molecular formula of tolypyridone D should be C15H21NO3. Compound 12 was determined to have a molecular weight of 278.1408 ([M-H]-) by high-resolution mass spectrometry, with a molecular formula of C15H21NO4. The two compounds have a mass difference of 16, which corresponds to an oxygen atom. Compound 12 should have an additional hydroxyl group compared to tolypyridone D, rather than an additional methyl group, as indicated in our labeling error, which has now been corrected.
Comment 2. The chemical structures in Figure 7 should be checked again. Terminal methyl group in compound 11 is missing. Please also check the other structures.
Response:
The chemical structures in Figure 7 have been checked.
Reviewer 3 Report
I appreciate the efforts by the authors to improve the manuscript. The writing has been visibly improved, discussion has received more attention (although not too much) and most points have been reasonably responded. But there are still some aspects to be improved or clarified before publication.
1. In relation to comment 1, in the sentence "... low expression levels under laboratory fermentation conditions" it should say "... low expression levels under our laboratory fermentation conditions". The authors should avoid the use of their fermentation conditions as a standard one in fungi.
2. In comment 5.1, I do not understand the explanation provided for the staircase. I do not see the logic of the connection with the band intensity. However, although I am not an expert in this technique, giving it more thought I find a possible explanation: depending on the goodness of the consensus sequence, it might happen that TopC binds and releases in an equilibrium that may result in a partial delay of the band, because the protein is not permanently bound to the DNA fragment. However, due to equilibrium, the higher the concentration of TopC, the longer the DNA fragment will have a TopC protein bound. Actually, with this explanation a blurring of the band could be expected, and this seems to be the case. So, the results with C and A-H promoters could be explained by a less efficient consensus element in these promoters, but it would fit with TopC binding.
So, I suggest the authors to change the interpretation in the text. This sentence:
"Notably, when binding to promoters designated as C or A-H, a staircase emerged, characterized by a progressive increase in size. Upon reviewing the chart, it becomes apparent that the binding experiments using C or AH probes exhibit lighter bands in comparison to those involving DE or FG probes. Consequently, we infer that this disparity is the predominant cause of the staircase."
Could be replaced by a simpler text, such as this:
"Notably, when binding to promoters designated as C or A-H, a staircase emerged, characterized by a progressive increase in size, suggesting a looser binding of the TopC protein"
3. In comment 5 it is said that "The blank control probe we used is the promoter of a un-specific gene". This is not explained in the results, but in the material and methods section it says: "as a negative control, 1 μg of DNA amplified from T. ophioglossoides genomic DNA was employed". This information is incomplete. What is amplified? The use of the negative control should be better explained.
4. Figure 7 has been improved, but the added compound names are too small to be properly read. Actually, labels are also too small in Figure 2 and 5 for enzyme names, and in Figure 5 also for other labels, that could be enlarged to some extent to improve clarity
Minor comments:
- Points are never used at the end of a article title
- For coherence, section 2.7 of Material and methods should be after section 2.5. So, sections 2.6 and 2.7 should be interchanged. Actually, the molecular biology sections and the chemical analytical sections should be contiguous in the M&M section.
- E. coli and A. tumefaciens names should be in italics in line 113
English has been considerably improved in this new version, but it still requires some minor revision. Some examples:
- the sentence " in lines 80-81 ... in numerous secondary metabolic biosynthesis" should be " ... in numerous secondary metabolic biosyntheses" or " ... in numerous secondary metabolite biosyntheses"
The sentece in lines 205-206 "Based on the analysis of the genome sequence of T. ophioglossoides for all gene clusters and found a gene cluster containing a PKS-NRPS hybrid enzyme." should be "Based on the analysis of the genome sequence of T. ophioglossoides for all gene clusters we found a gene cluster containing a PKS-NRPS hybrid enzyme." or "The analysis of the genome sequence of T. ophioglossoides for all gene clusters led us to find a gene cluster containing a PKS-NRPS hybrid enzyme."
- The sentence in lines 216-217 “This gene encodes a fungal cluster-specific C6 transcriptional factor. (Figure 1), likely plays a role in regulating the expression of cluster genes.” Should be “This gene encodes a fungal cluster-specific C6 transcriptional factor (Figure 1), which likely plays a role in regulating the expression of cluster genes.”
- I do not think “molecular formular” in lines 282, 285, 289 y 290 is correct
Author Response
Review 3:
Comment 1. 1. In relation to comment 1, in the sentence "... low expression levels under laboratory fermentation conditions" it should say "... low expression levels under our laboratory fermentation conditions". The authors should avoid the use of their fermentation conditions as a standard one in fungi.
Response: Thank you very much for your suggestions, we have made the relevant modifications. We have changed "low expression levels under laboratory fermentation conditions" to "... low expression levels under our laboratory fermentation conditions."
Comment 2. 2. In comment 5.1, I do not understand the explanation provided for the staircase. I do not see the logic of the connection with the band intensity. However, although I am not an expert in this technique, giving it more thought I find a possible explanation: depending on the goodness of the consensus sequence, it might happen that TopC binds and releases in an equilibrium that may result in a partial delay of the band, because the protein is not permanently bound to the DNA fragment. However, due to equilibrium, the higher the concentration of TopC, the longer the DNA fragment will have a TopC protein bound. Actually, with this explanation a blurring of the band could be expected, and this seems to be the case. So, the results with C and A-H promoters could be explained by a less efficient consensus element in these promoters, but it would fit with TopC binding.
So, I suggest the authors to change the interpretation in the text. This sentence:
"Notably, when binding to promoters designated as C or A-H, a staircase emerged, characterized by a progressive increase in size. Upon reviewing the chart, it becomes apparent that the binding experiments using C or AH probes exhibit lighter bands in comparison to those involving DE or FG probes. Consequently, we infer that this disparity is the predominant cause of the staircase."
Could be replaced by a simpler text, such as this:
"Notably, when binding to promoters designated as C or A-H, a staircase emerged, characterized by a progressive increase in size, suggesting a looser binding of the TopC protein"
Response: We have made the relevant modifications following your advice.
Comment 3. In comment 5 it is said that "The blank control probe we used is the promoter of an un-specific gene". This is not explained in the results, but in the material and methods section it says: "as a negative control, 1 μg of DNA amplified from T. ophioglossoides genomic DNA was employed". This information is incomplete. What is amplified? The use of the negative control should be better explained.
Response: We have added the relevant information in both the Materials and Methods and Results sections.
Comment 4. Figure 7 has been improved, but the added compound names are too small to be properly read. Actually, labels are also too small in Figure 2 and 5 for enzyme names, and in Figure 5 also for other labels, that could be enlarged to some extent to improve clarity
Response: Thank you very much for your advice. We have made the relevant modifications as suggested.
Minor comments:
- Points are never used at the end of a article title
- For coherence, section 2.7 of Material and methods should be after section 2.5. So, sections 2.6 and 2.7 should be interchanged. Actually, the molecular biology sections and the chemical analytical sections should be contiguous in the M&M section.
- E. coli and A. tumefaciens names should be in italics in line 113
Response: Thank you very much for your advice. We have made the relevant modifications as suggested.